# Tuning social interactions' strength drives collective response to light intensity in schooling fish

Tingting Xue[1,2☺], Xu Li[1,2☺], GuoZheng Lin[1,2], Ramón Escobedo[2], Zhangang Han[1], Xiaosong Chen[1], Clément Sire[3], Guy Theraulaz[2]*

**1** School of Systems Science, Beijing Normal University, Beijing, China, **2** Centre de Recherches sur la Cognition Animale, Centre de Biologie Intégrative (CBI), CNRS & Université de Toulouse III – Paul Sabatier, Toulouse, France, **3** Laboratoire de Physique Théorique, CNRS & Université de Toulouse III – Paul Sabatier, Toulouse, France

☺ These authors contributed equally to this work.
* guy.theraulaz@univ-tlse3.fr

**Data Availability Statement:** All data needed to evaluate the conclusions in the paper are present in the paper and/or the Supplementary Materials or

## Abstract

Schooling fish heavily rely on visual cues to interact with neighbors and avoid obstacles. The availability of sensory information is influenced by environmental conditions and changes in the physical environment that can alter the sensory environment of the fish, which in turn affects individual and group movements. In this study, we combine experiments and data-driven modeling to investigate the impact of varying levels of light intensity on social interactions and collective behavior in rummy-nose tetra fish. The trajectories of single fish and groups of fish swimming in a tank under different lighting conditions were analyzed to quantify their movements and spatial distribution. Interaction functions between two individuals and the fish interaction with the tank wall were reconstructed and modeled for each light condition. Our results demonstrate that light intensity strongly modulates social interactions between fish and their reactions to obstacles, which then impact collective motion patterns that emerge at the group level.

## Author summary

Schooling fish rely extensively on visual cues to interact with their peers and navigate obstacles. Environmental conditions can modify the sensory landscape experienced by fish, and in turn impact both individual and collective movements. Here, we combine experiments and data-driven modeling to explore the influence of different levels of light intensity on social interactions and collective behavior in rummy-nose tetra. By reconstructing and modeling the interactions between pairs of fish and between fish and the tank boundary, we show that light intensity modulates social interactions and influences how fish swim and respond to obstacles. Our model explains how the modulation of these interactions at the individual level leads to changes in collective movements observed at the group level.

available at the following online repository: https://doi.org/10.6084/m9.figshare.22640053.v1.

**Funding:** T.X. received a grant from the China Scholarship Council (CSC N°202106040094). X.L. received a grant from the China Scholarship Council (CSC N°202106040093). G.L. received a grant from the China Scholarship Council (CSC N° 202006040162). Z.H. and T.X. were supported by the National Natural Science Foundation of China under Grant N° 62176022. X.C. and X.L. were supported by the National Natural Science Foundation of China under Grant N°12135003. G. T., R.E. and C.S. were supported by the Agence Nationale de la Recherche (ANR-20-CE45-0006-1). G.T. acknowledges the support of the Indo-French Centre for the Promotion of Advanced Research (project N° 64T4-B). G.T. also gratefully acknowledges the Indian Institute of Science to serve as Infosys visiting professor at the Centre for Ecological Sciences in Bengaluru. The funders had no role in study design, data collection and analysis, decision to publish, or preparation of the manuscript.

**Competing interests:** The authors have declared that no competing interests exist.

## Introduction

Collective behaviors are observed across multiple spatial scales in nature, as seen in bacteria colonies, insects, bird flocks, or fish shoals [1–3]. By enabling individuals to coordinate their actions, these phenomena bear significant functional consequences for group members, including improved safety [4–7], increased foraging [4, 5, 8], and enhanced reproductive success [3, 9]. It is widely accepted that collective behaviors emerge from the interactions between individuals within a group [1, 2, 7, 10–12]. These social interactions have fundamental effects on the phenotypes and fitness of individuals, as well as the collective behavior of groups [13–15]. They can even change the activity patterns of individuals to influence the fitness of others [16]. Analyzing social interactions among individuals is thus a key factor in understanding and controlling the mechanisms of collective animal behavior [1–3, 17–20].

In recent decades, the analysis of collective behavior has progressed in terms of understanding individual interactions [1, 3, 21, 22]. Advances in computerized methods based on learning algorithms have overcome the difficulty of automatically tracking groups of animals, enabling the quantitative study of the effects of social interactions on individual behaviors [23–26]. These methods have been applied to various biological systems, from schools of fish [27–29] and flocks of birds [30–32], to groups of primates [33, 34] and human crowds [35, 36], providing new directions for quantifying collective motion. Animal experiments have successfully linked group-level functional properties to behavioral mechanisms at the individual scale, while also accurately obtaining large amounts of data about individuals in a group [37–39]. Using these tracking data, one can now reconstruct and model the social interactions between individuals, as well as their interactions with obstacles present in the environment, to predict the properties of the collective motion [39–42]. Our computational model allows to test, quantify, and interpret the impact of visual cues on individual and collective motion.

In fish schools, social interactions rely on the integration of multiple sensory stimuli [43, 44] including vision [45] and lateral line [46, 47], which are used to detect movements of neighbors and vibrations of the surrounding water. Generally, the availability of sensory information is modulated by environmental conditions, and changes in the physical environment can alter the sensory environment of animals, which in turn affects individual and group movements. Previous studies have shown how various environmental factors, including turbidity, oxygen levels, and light levels, affect the collective behavior of fish [48–51]. For instance, turbid water scatters and reduces the amount of light, which can even cause changes in the spectrum, resulting in fish having less access to public information and opportunities for social learning about food locations under turbid conditions [52, 53]. Interestingly, in order to counteract this limitation of a decrease in information exchange in highly turbid water, juvenile cod (*Gadus morhua*) keep their foraging rates constant by increasing their activity levels [54]. Hypoxia can affect the school structure and dynamics of the fish, as well as cause an increase in school volume and eventually lead to the school breaking down [55]. In many species, the diurnal dynamics of illumination is responsible for the school disintegration and the loss of schooling observed at night [56]. Artificial light at night can also affect the activity patterns of individuals, dramatically changing the nature of their motion [57]. The study of individual behavior and social interactions under these different environmental factors is an important step to understand the adaptive capabilities and the ecological success of a species.

Here, we used a combination of experiments with faithful data-based modeling to investigate the impact of varying levels of light intensity on social interactions and the resulting collective behavior in rummy-nose tetra fish (*Hemigrammus rhodostomus*). The Rummy-

nose tetra is a species that has a strong tendency to school, with an intermittent swimming mode characterized by alternating bursts and coasting phases. This swimming mode allows us to analyze individual trajectories as a series of discrete behavioral decisions in time and space [39].

Previous studies have already investigated the role of lighting on the dynamics of collective swimming in rummy-nose tetra [43, 58]. However, they did not analyze in this context the behavioral mechanisms and social interactions at play at the individual scale in small and large groups. Our experimental and simulation results indicate that the level of illumination does not modify the general form of social interactions between fish and their interaction with the tank wall, but only modulates the intensity and range of these interactions. Ultimately, our computational approach makes it possible to establish a direct causal link between (1) the modulation of these interactions by light intensity at the individual scale and (2) the specific collective motion patterns that emerge at the collective level in groups of different sizes.

## Results

The "burst-and-coast" swimming mode of *H. rhodostomus* consists of the successive alternation of sudden accelerations and quasi-passive deceleration periods during which the fish glides along a near straight line. Each acceleration-deceleration sequence is called a "kick". Changes of direction in fish motion take place at the onset of a kick, during the acceleration phase (see S1 Fig).

### Effect of light intensity on individual swimming behavior

Light intensity deeply affects fish behavior and its burst-and-coast swimming (Fig 1A, 1B, and S1 Video). We find that the average kick duration $\langle \tau \rangle$ and the average kick length $\langle l \rangle$ increase with light intensity, with $\langle \tau \rangle = 0.31 \pm 0.01$ s at 0.5 lx *vs* $\langle \tau \rangle = 0.51 \pm 0.02$ s at 50 lx, and $\langle l \rangle = 31 \pm 2$ mm at 0.5 lx *vs* $\langle l \rangle = 62 \pm 4$ mm at 50 lx (Fig 2A, 2B, 2D, 2E, and S9 Table). However, the average peak speed $v_0$ does not significantly vary with light intensity (Fig 2C and 2F). We have performed a Wilcoxon rank-sum test for $\tau$, $l$, $v_0$ (see S9 Table; similar tests are presented in S10–S12 Tables, for groups of $N$ = 2, 5, 25 fish), which provides a statistical justification of our claim that $\tau$ and $l$ are consistently increasing with the light intensity before saturating for 5–50 lx, whereas $v_0$ does not exhibit any systematic trend.

During the gliding phase, fish swims in a straight line with an exponentially decaying speed. We find that the speed decays the fastest when light intensity is the lowest (0.5 lx) with a corresponding decay rate $\tau_0 = 0.34$ s (S2(A) Fig). In addition, we find that, when the fish is far from the wall ($r_w > 60$ mm, a distance for which the influence of the wall becomes negligible), the probability density function (PDF) of the spontaneous heading fluctuations $\delta\phi_R$ is Gaussian (S3 Fig).

### Effect of light intensity on the interaction of fish with the wall

Due to symmetry constraints in a circular tank, the interaction between a fish and the wall can only depend on its distance to the wall $r_w$ and its relative angle with the wall $\theta_w$ (Fig 1C). Fig 3A shows the experimental PDF of $r_w$ for different light intensities, illustrating that the higher the light intensity, the closer the fish is to the wall: $\langle r_w \rangle = 33 \pm 3$ mm at 0.5 lx *vs* $\langle r_w \rangle = 20 \pm 2$ mm at 50 lx. Fig 3B shows the PDF of $\theta_w$, centered near but below 90˚, indicating that the fish generally stays almost parallel to the wall while frequently swimming towards it.

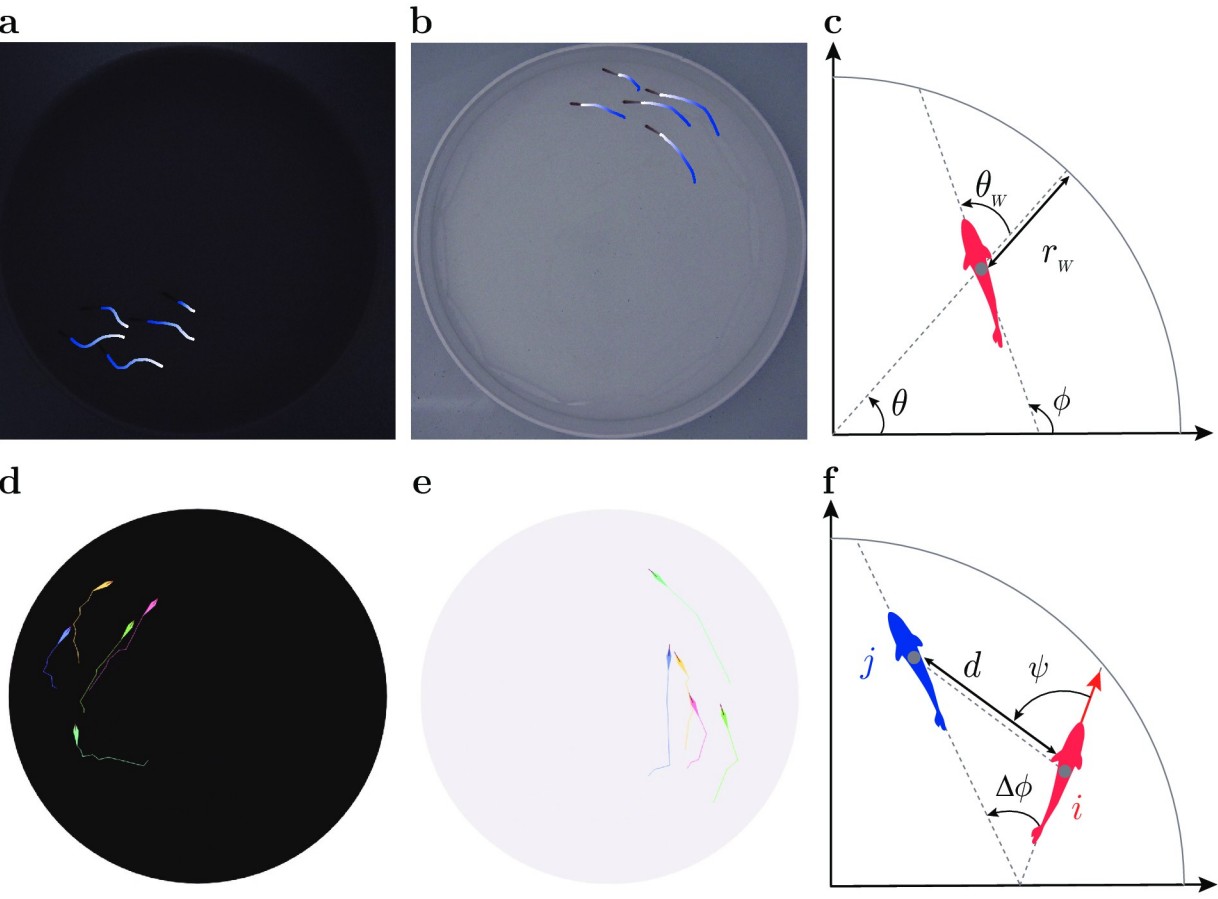

**Fig 1. Collective motion in groups of fish under different light conditions. a,b** Trajectories of 5 fish swimming in a tank during the experiments at low light intensity (0.5 lx) and high light intensity (50 lx), respectively. The trajectories show the successive positions of individuals over the past 1 s. **c** State variables of the fish with respect to the tank, position angle $\theta$ and fish heading angle $\phi$, and with respect to the wall, distance $r_w$ and relative orientation $\theta_w$. **d,e** Numerical simulations of the model for $N = 5$ in low light and high light conditions respectively. **f** State variables of a focal fish (red) with respect to a neighbor (blue): distance between them $d$, viewing angle $\psi$, and relative orientation $\Delta\phi$.

## Modeling and measurement of fish interaction with the wall in different light conditions

To measure experimentally the interactions between a fish and the wall, we use the procedure introduced by Calovi *et al.* [39]. The result of this procedure is presented as a scatter plot in Fig 4A and 4B along with the simple functional forms $f_w(r_w)$ and $O_w(\theta_w)$ used to fit these data (Eqs (9) and (10) in Material and methods). We find that the shape of the repulsive form is the same for all light intensities (Fig 4). Fig 5A shows that the intensity of the spontaneous heading fluctuation $\delta\phi_R$ increases with light intensity until reaching a plateau around 5 lx, where $\delta\phi_R \approx$ 0.35. Moreover, both the effective strength $\gamma_w$ and range $l_w$ of the interaction with the wall increase with light intensity (Fig 5B and 5C), while the angular dependence of the interaction remains almost unchanged (Figs 4B and S4). For high light intensity, there are significant deviations between the fits of $f_w(r_w)$ and $O_w(\theta_w)$ and the actual data points, which indirectly confirms that individual movement patterns are indeed different under varying light intensities. Fish prefer to stay closer to the walls and to move in directions parallel to the wall under high light intensity (see Fig 3). This leads to a concentration of the data distribution, with relatively few data points available when fish are far from the wall and non-parallel to the wall.

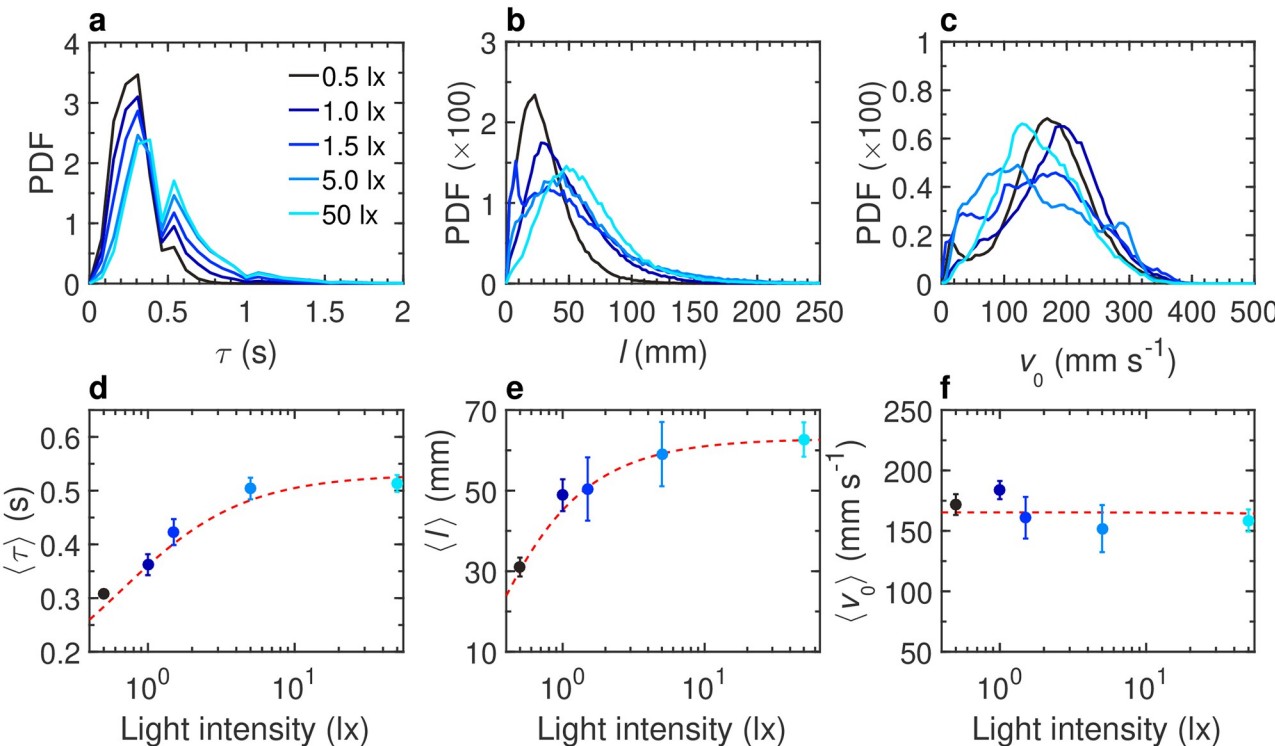

**Fig 2. Effects of light intensity on the burst-and-coast swimming of a single fish. a** Probability density function (PDF) of the duration between two consecutive kicks $\tau$, **b** PDF of the distance traveled by a fish between two kicks $l$, **c** PDF of the maximum speed when the fish performs a kick $v_0$, at different light intensities: 0.5, 1, 1.5, 5, and 50 lx (from dark to light blue). **d** Average duration between two consecutive kicks $\langle\tau\rangle$, **e** average distance traveled by a fish between two kicks $\langle l\rangle$, and **f** average speed when the fish performs a kick $\langle v_0\rangle$, as functions of light intensity. Solid circles are the average values on all experiments; error bars represent the standard error. Red dashed lines show the trend of the average value with the light intensity.

Consequently, the reconstruction of the interaction with the wall is reasonably precise in the regions of $r_w$ and $\theta_w$ associated to a high probability, but exhibits large fluctuations for $r_w > 80$ mm or $\theta_w$ far enough from ±90˚.

Altogether, these results show that the repulsion exerted by the wall on fish increases with light intensity. This is a direct consequence of the modulation of the visual perception of fish by the level of illumination (see also S5 Fig).

We then implement the interaction of the fish with the wall in the burst-and-coast model (see Material and methods section). In Fig 3, and for the 5 light intensities considered, we compare the distribution of the distance to the wall $r_w$ and the relative angle of the fish with the wall $\theta_w$, as obtained experimentally and in extensive numerical simulations of the model, finding an overall satisfactory agreement. The numerical values of the parameters used in the model are listed in S5 Table. On a more qualitative note, the simulations of the model reproduce fairly well the behavior and motion of a real fish under the different light conditions (Fig 1D, 1E, and S5 Video).

## Effect of light intensity on social interactions between two fish

When fish swim in pairs, both the average kick length and kick duration increase with light intensity: $\langle l\rangle = 23 \pm 2$ mm at 0.5 lx *vs* $\langle l\rangle = 50 \pm 4$ mm at 50 lx, and $\langle\tau\rangle = 0.31 \pm 0.01$ s at 0.5 lx *vs* $\langle\tau\rangle = 0.62 \pm 0.02$ s at 50 lx (Figs 6 and S6 and S10 Tables and S2 Video). Fig 7A and 7B shows the PDF of the distance to the wall of both fish when we distinguish them by their

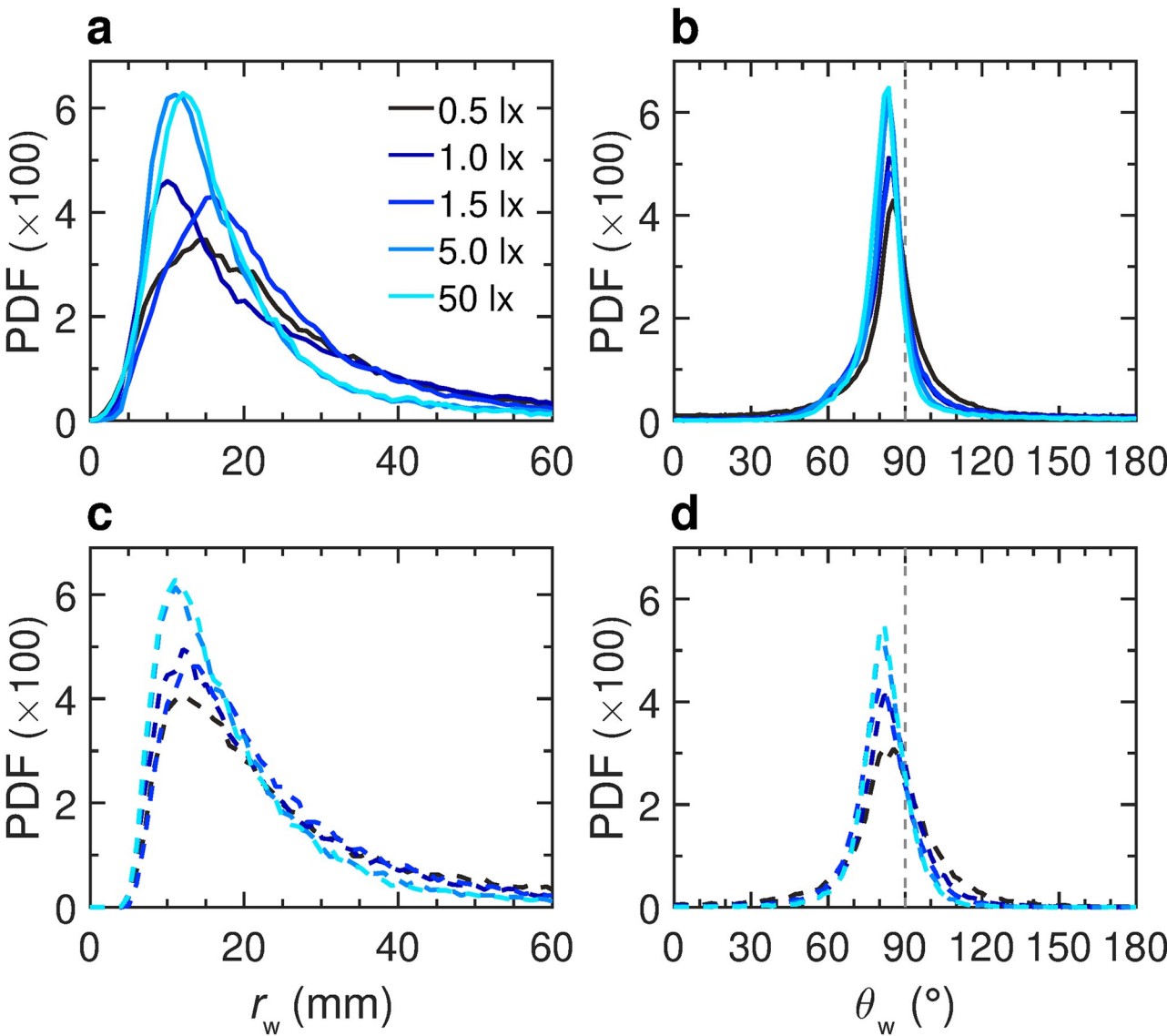

**Fig 3. Effects of light intensity on the spatial distribution and motion of a fish swimming alone. a,b** Probability density functions (PDF) of the distance to the wall $r_w$ and the relative angle of the fish with the wall $\theta_w$, respectively, measured in the experiments for the 5 light intensities: 0.5, 1, 1.5, 5, and 50 lx (solid lines, from dark to light blue). **c,d** PDFs of $r_w$ and $\theta_w$, respectively, in the numerical simulations of the model and for each light condition (dashed lines).

instantaneous relative position as follows: the geometrical leader is defined as the fish with the largest viewing angle of the other fish $|\psi|$, that is, the fish which needs to turn the most to directly face the other fish, the other fish being therefore the geometrical follower (Fig 1F) [39]. We find that the geometrical leader is closer to the wall than the follower, as already noted in [39], but both fish are further away from the wall than an isolated fish, although this second effect is less pronounced at the maximum illumination ($\langle r_w^{\text{leader}} \rangle = 63 \pm 1$ mm and $\langle r_w^{\text{follower}} \rangle = 70 \pm 1$ mm at 0.5 lx $vs$ $\langle r_w^{\text{leader}} \rangle = 41 \pm 1$ mm and $\langle r_w^{\text{follower}} \rangle = 53 \pm 1$ mm at 50 lx). In fact, the follower fish is taking a shortcut through the circular tank to catch up with the leader, resulting in the follower swimming farther away from the wall of the tank, and also slightly attracting the leader away from the wall. Fig 7C and 7F shows the PDF of the relative

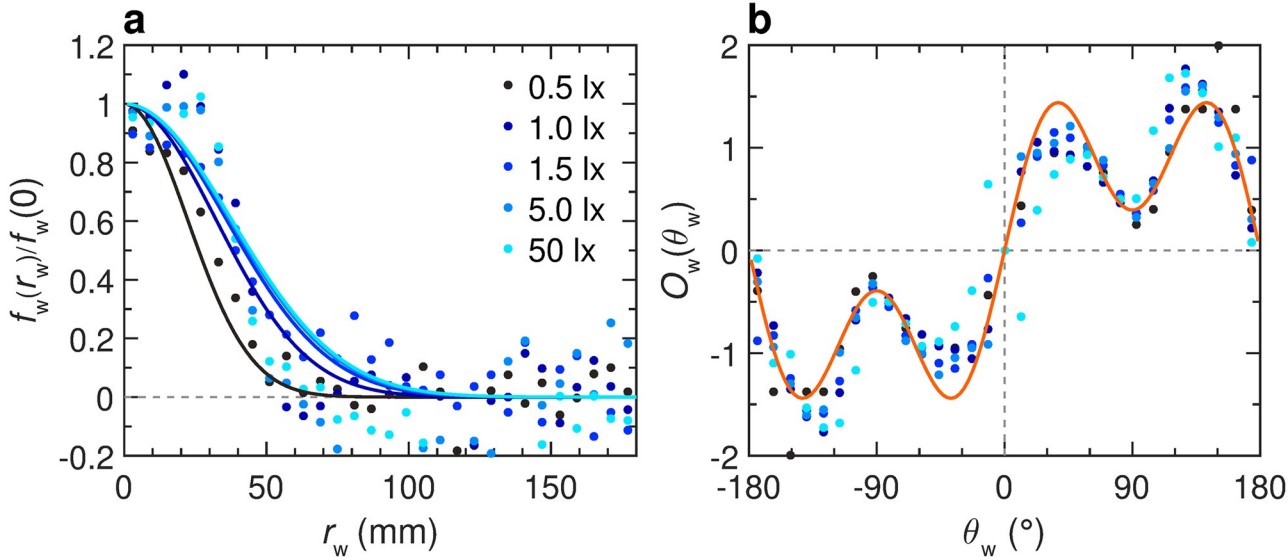

**Fig 4. Effects of light intensity on the fish interactions with the tank wall.** Function of repulsion $f_w(r_w)O_w(\theta_w)$ as extracted from the experiments by means of the reconstruction procedure, for different light intensities: 0.5, 1, 1.5, 5, and 50 lx (from dark to light blue). **a** Intensity $f_w(r_w)$ of the interaction as a function of the fish distance to the wall $r_w$ (see Fig 3) has only a small residual weight of 8.1%, 4.2%, 3.6%, 1.8%, 1.3% for $r_w > 80$ mm, respectively, which explains the large fluctuations of $f_w(r_w)$ observed for $r_w > 80$ mm. **b** Intensity $O_w(\theta_w)$ of the interaction as a function of the relative orientation of the fish to the wall $\theta_w$. Color dots correspond to the discrete values resulting from the reconstruction procedure, extracted from the experimental data. Blue solid lines correspond to the analytical approximation of the discrete functions for the corresponding light condition. The orange line corresponds to the analytical approximation of a single discrete function combining all light conditions: $O_w(\theta_w) = 1.9612 \sin(\theta_w)[1 + 0.8 \cos(2\theta_w)]$.

wall angle $\theta_w$ of the geometrical leader and follower respectively, which are also wider than for a single fish. Fig 7I shows that the distance $d$ between the two fish decreases when light intensity increases, suggesting that attraction between fish increases with light intensity (see next section): $\langle d \rangle = 76 \pm 9$ mm at 0.5 lx vs $\langle d \rangle = 67 \pm 7$ mm at 50 lx. Moreover, the PDF of the viewing angle $\psi$ (see Fig 1F) of the leader and follower show a marginal variation with light (Fig 7J and 7K), except at the highest illumination where the leader appears to swim more in front of the follower, which is consistent with the fact that both fish swim closer to the wall at this

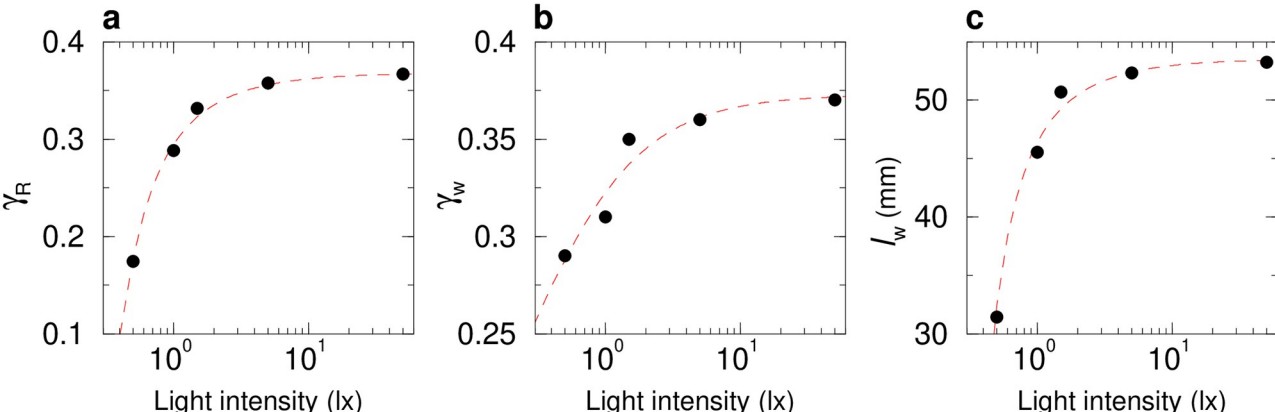

**Fig 5. Effects of light intensity on the parameters of the fish interactions with the tank wall. a** Spontaneous heading fluctuations of a fish, $\gamma_R$, as a function of light intensity when the fish is far from the tank wall ($r_w > 60$ mm). **b** Intensity of the wall repulsion, $\gamma_w$, as a function of light intensity. **c** Range of the the wall repulsion, $l_w$, as a function of light intensity. Red dashed lines show the trend of the average value with light intensity.

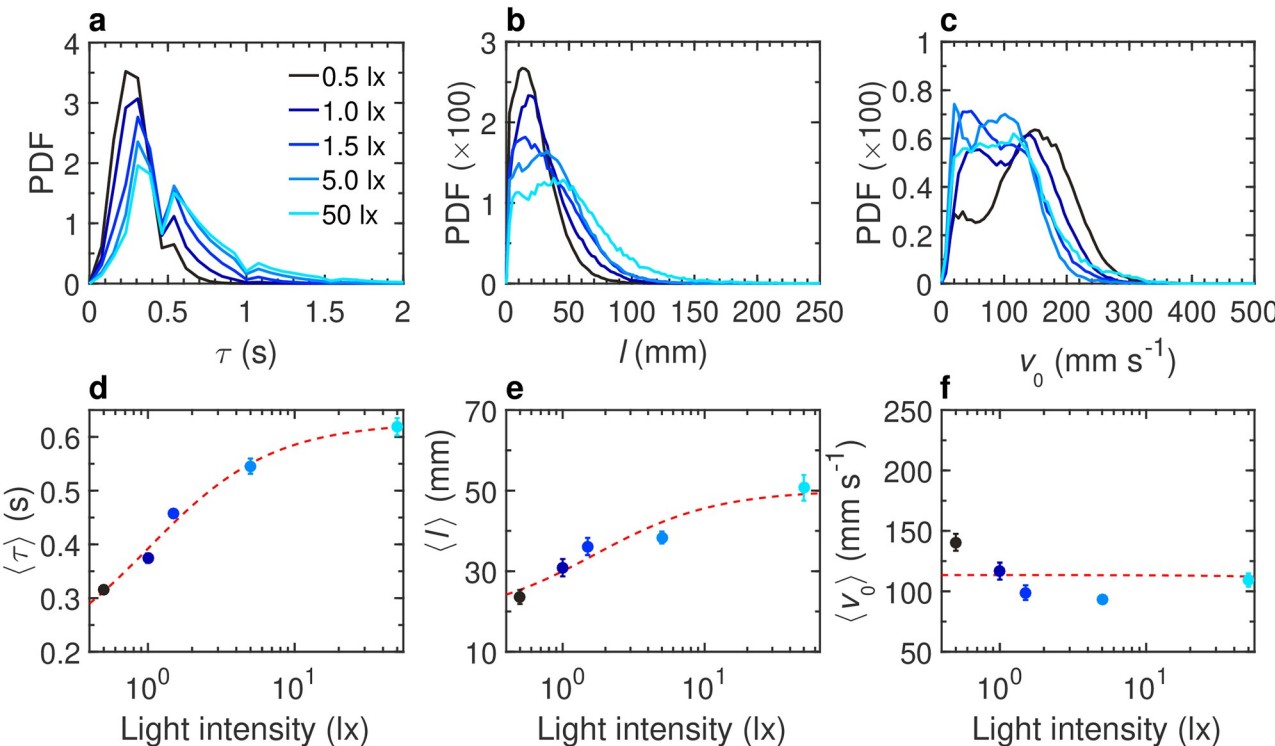

**Fig 6. Effects of light intensity on the burst-and-coast swimming of pairs of fish. a** Probability density function (PDF) of the duration between two consecutive kicks $\tau$, **b** PDF of the distance traveled by a fish between two kicks $l$, **c** PDF of the maximum speed when the fish performs a kick $v_0$, at different light intensities: 0.5, 1, 1.5, 5, and 50 lx (from dark to light blue). **d** Average duration between two consecutive kicks $\langle \tau \rangle$, **e** average distance traveled by a fish between two kicks $\langle l \rangle$, and **f** average speed when the fish performs a kick $\langle v_0 \rangle$, as functions of light intensity. Solid circles are the average values on all experiments; error bars represent the standard error. Red dashed lines show the trend of the average value with the light intensity.

maximum illumination. Finally, The PDF of their relative orientation $\Delta\phi$ (Fig 7L) is essentially not affected by the light intensity.

## Modeling and measurement of social interactions between fish in different light conditions

As shown in Calovi *et al.* [39], social interactions between fish in *H. rhodostomus* combine attraction and alignment. We assume that both the attraction and alignment interactions $F_{\text{Att}}$ and $F_{\text{Ali}}$ can be decomposed into the decoupled product of three functions, each one depending on one of the three variables that determine the relative states of the two fish, namely, the distance between fish $d$, their relative orientation $\Delta\phi$, and the viewing angle $\psi$ with which the focal fish perceives its neighbor (Fig 1F). We extract the analytical expressions reproducing the main features of the attraction and alignment interactions from the experimental data by means of the reconstruction procedure developed in [39]. The dots on Fig 8 show the results of the extraction of the interaction functions from experimental data, and the solid lines are the simple functional forms used to fit these data (see Eqs (14)–(19) in Section Computational model). Figs 8A, 8D and 9 show that the strength and the range of both attraction and alignment increase with light intensity. S7 and S8 Figs show these functions in more detail. Altogether, these results suggest that visual information is a key factor in the ability of fish to coordinate their collective swimming, in line with previous works [59]. As the perception of the position of its neighbor by a fish becomes more precise, the intensity of social interactions

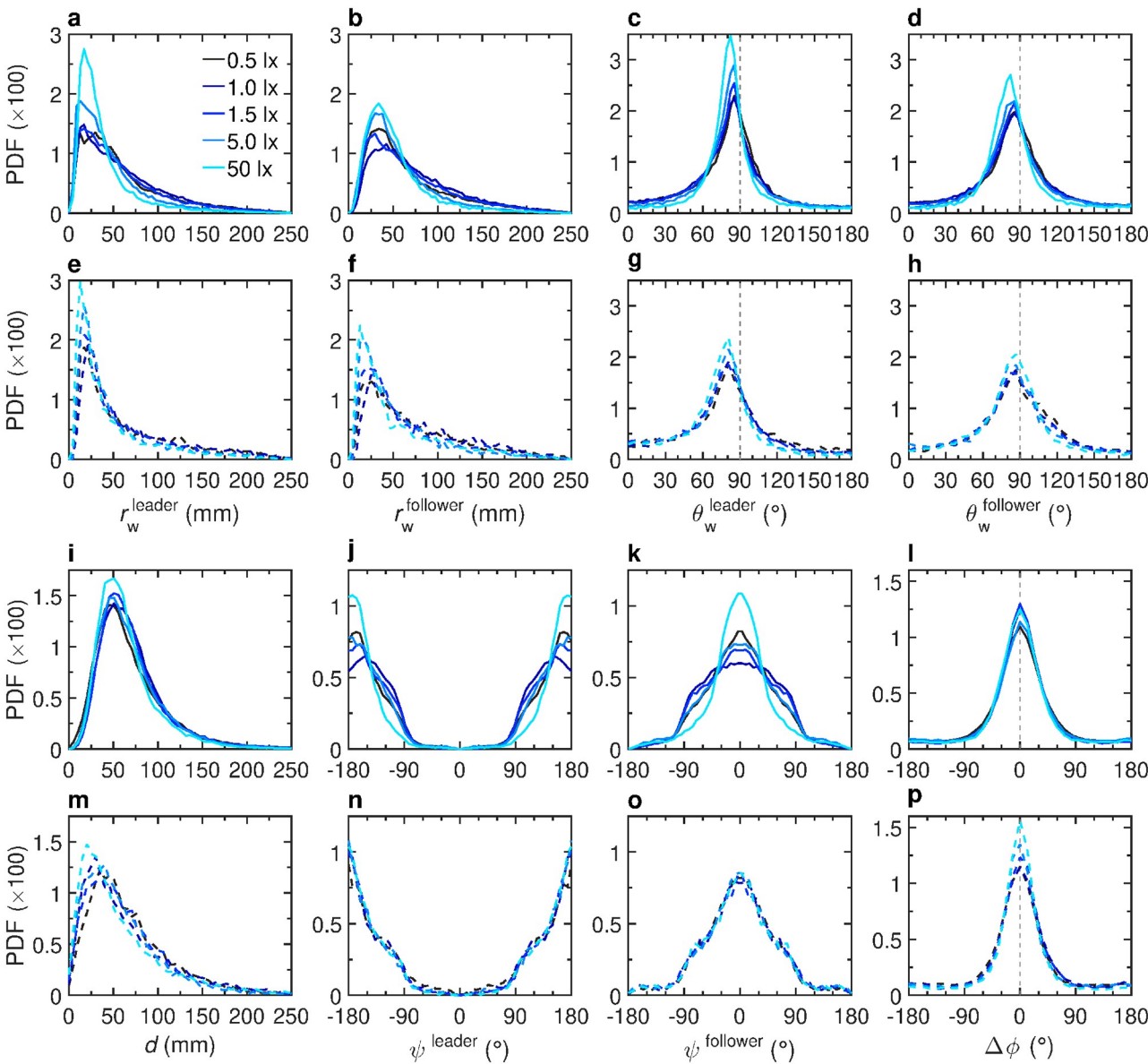

**Fig 7. Effects of light intensity on the spatial distribution in groups of two fish.** Probability density functions (PDF) of **a,e** the distance to the wall $r_w$ of the geometrical leader, **b,f** the distance to the wall $r_w$ of the geometrical follower, **c,g** the relative angle to the wall $\theta_w$ of the geometrical leader, **d,h** the relative angle to the wall $\theta_w$ of the geometrical follower, **i,m** the distance between the two fish $d$, **j,n** the viewing angle $\psi$ of the geometrical leader, **k,o** the viewing angle $\psi$ of the geometrical follower, and **l,p** the relative orientation $\Delta\phi$ between the two fish, for 5 different light intensities 0.5, 1, 1.5, 5, and 50 lx (from dark to light blue). Solid lines (**a-d,i-l**) correspond to the experimental results and dashed lines (**e-h,m-p**) correspond to numerical simulations of the model.

becomes stronger, which leads the fish to be closer to each other. However, the angular dependence of both attraction (Fig 8B and 8C) and alignment (Fig 8E and 8F) are not affected by light intensity. We then introduced the functional forms of Eqs (14)–(19) that adequately describe $F_{Att}(d, \Delta\phi, \psi)$ and $F_{Ali}(d, \Delta\phi, \psi)$ in the model to simulate the motion of fish (see Material and methods; the parameter values used in the simulations are given in S6 Table).

Fig 7E-7H and Fig 7M-7P show the results of extensive numerical simulations of the model including the interactions between fish, compared to the experimental measures above

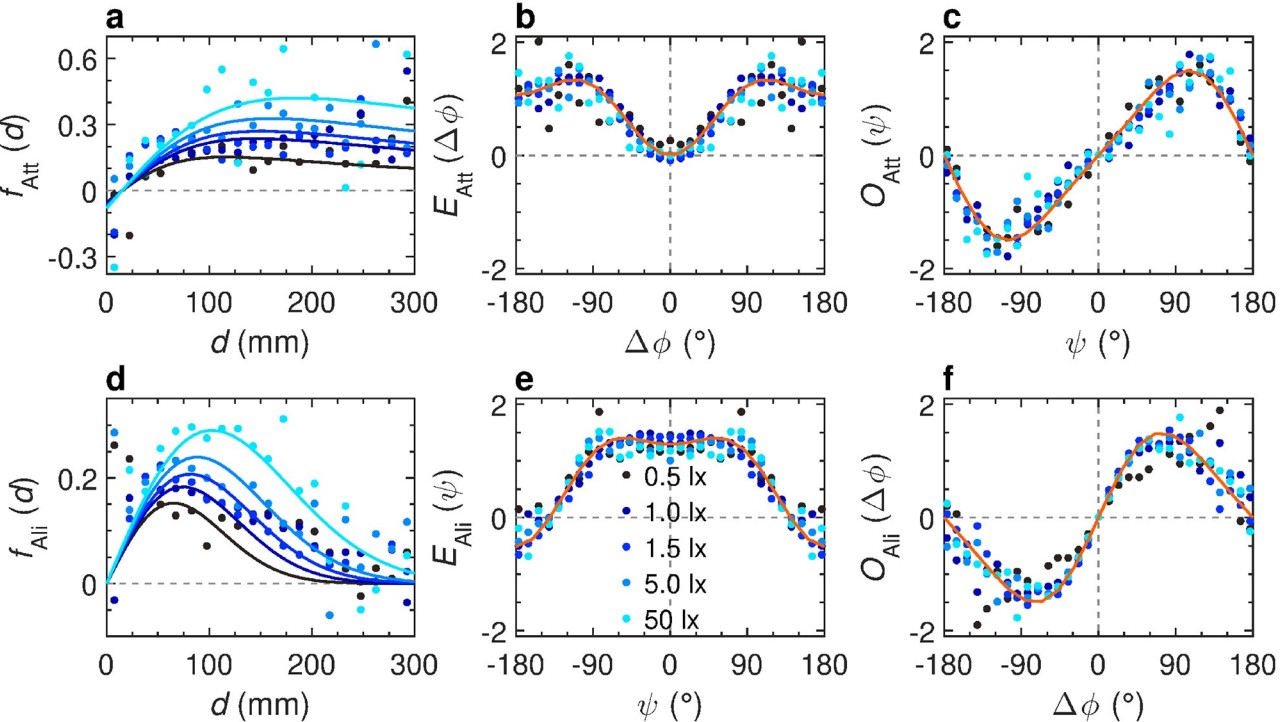

**Fig 8. Effects of light intensity on social interactions between pairs of fish. a-c** Attraction interaction function $F_{Att} = f_{Att}(d)O_{Att}(\psi)E_{Att}(\Delta\phi)$ and **d-f** alignment interaction function $F_{Ali} = f_{Ali}(d)O_{Ali}(\Delta\phi)E_{Ali}(\psi)$, for 5 different light intensities 0.5, 1, 1.5, 5, and 50 lx (from dark to light blue). **a,d** Intensity of attraction and alignment respectively as functions of the distance between fish $d$. **b,c** Even and odd modulations of attraction intensity as functions of the relative orientation $\Delta\phi$ and the viewing angle $\psi$ respectively. **e,f** Even and odd modulations of alignment intensity as functions of $\psi$ and $\Delta\phi$ respectively. Color dots correspond to the discrete values resulting from the reconstruction procedure, extracted from the experimental data. Blue solid lines correspond to the analytical approximation of the discrete functions for the corresponding light condition. Orange lines correspond to the analytical approximation of a single discrete function combining all light conditions.

described. Overall, we find a qualitative (S6 Video) and fair quantitative agreement. Fig 9A-9D show in detail the trend, as light intensity increases, of the numerical values of the strength and range parameters of both interactions $\gamma_{Att}$, $l_{Att}$, $\gamma_{Ali}$, and $l_{Ali}$ respectively that we used in the simulations. In the attraction interaction, both $\gamma_{Att}$ and $l_{Att}$ increase with light intensity. In the alignment interaction, $\gamma_{Ali}$ decreases with light intensity, but this is compensated by the increase of $l_{Ali}$, resulting in an interaction that is stronger when light is more intense (Fig 8D). These results show that all parameters converge to a saturation value as light becomes more intense.

## Effect of light intensity on collective behavior in groups of 5 and 25 fish

We then investigate the consequences of the modulation of the interaction strength by the light intensity on the collective behavior in groups of $N = 5$ and 25 fish (see S3 and S4 Videos). S10 and S11 Figs and S10 and S12 Tables show the effects of light intensity on the duration between two consecutive kicks $\tau$, the distance traveled by a fish between two kicks $l$, and the maximum speed when the fish performs a kick $v_0$, in both group sizes. We characterize the collective patterns by means of 5 observables quantifying the behavior and the spatial distribution of the school: 1) the distance of a fish to the tank wall, $r_w(t)$; 2) the distance of a fish to its nearest neighbor, $NND(t)$; 3) the group radius of gyration, $D(t)$, equal to the standard deviation of the distance of the $N$ fish to the barycenter of the group; 4) the group polarization, $P(t)$, which quantifies the mutual alignment of the fish; 5) the milling index, $M(t)$, which quantifies the

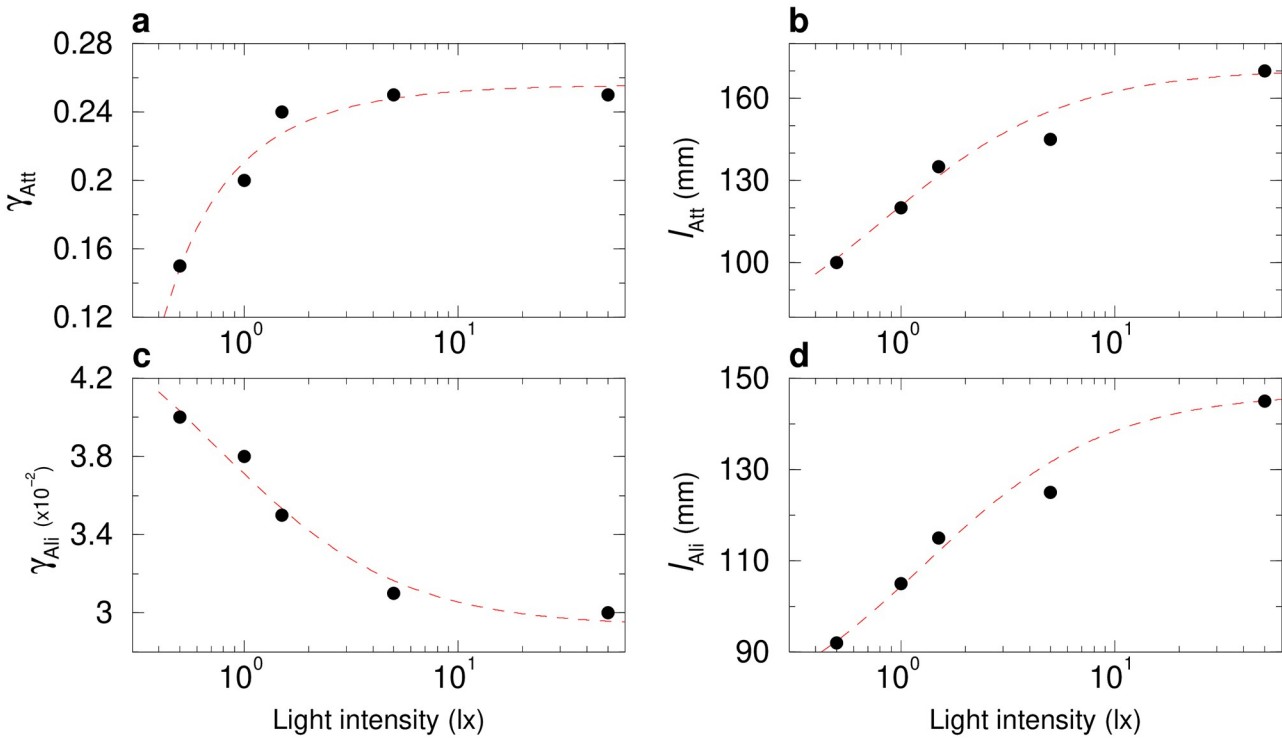

**Fig 9. Effect of light intensity on the strength and range interaction parameters for $N = 2$ fish.** Parameter values of **a** attraction strength $\gamma_{Att}$, **b** attraction range $l_{Att}$, **c** alignment strength $\gamma_{Ali}$, and **d** alignment range $\gamma_{Ali}$, used in the numerical simulations, as functions of light intensity (0.5, 1, 1.5, 5, and 50 lx). Dashed lines show the trend of the average value with light intensity.

global rotation of the fish around the center of the tank, in a vortex-like formation (see Material and methods).

Fig 10 shows the mean values of all these observables for all light conditions, and S12 and S13 Figs show the corresponding PDFs. Let us mention at this point that when the number of fish is increasing, the impact of the finite size of the confining tank on the group increases, which will be especially apparent for $N = 25$ fish.

As the number of fish increases, they swim farther from the wall (Fig 10A and 10F). As a function of light, we obtained a clear pattern for $N = 1$ and $N = 2$ (reproduced by the model), showing that the fish are swimming markedly closer to the wall as the illumination is increasing. For $N = 5$ and $N = 25$, this overall pattern is much less clear. For $N = 5$ and $N = 25$, the group is more compact than for $N = 2$, with a mean distance between nearest neighbors in the range 35–45 mm ($N = 5$) and 45–50 mm ($N = 25$), instead of 65–75 mm for $N = 2$ (Fig 10B and 10G). The fact that this mean distance is higher for $N = 25$ than for $N = 5$ reflects the fact that the groups of 25 fish are spread over the whole tank (see hereafter), although it remains denser near the wall. In the case $N = 2$, the mean distance between the 2 fish was consistently decreasing with increasing light, whereas, for $N = 5$ and $N = 25$, it slightly increases up to 1.5–5 lx before saturating or even slightly decreasing again at maximum illumination, although the variation range remains quite narrow. The radius of gyration of groups of $N = 5$ and $N = 25$ fish slightly increases for the lowest illumination before saturating (Fig 10C and 10H).

The main difference between groups of 5 and 25 fish is highlighted by their mean polarization and milling order parameters (Fig 10D, 10E, 10I, and 10J). The group of $N = 5$ fish is still well localized in the tank of radius $R = 250$ mm (with a mean radius of gyration $\langle D \rangle \approx$ 50 mm $\ll R$), and the good alignment between fish is reflected by a high mean polarization

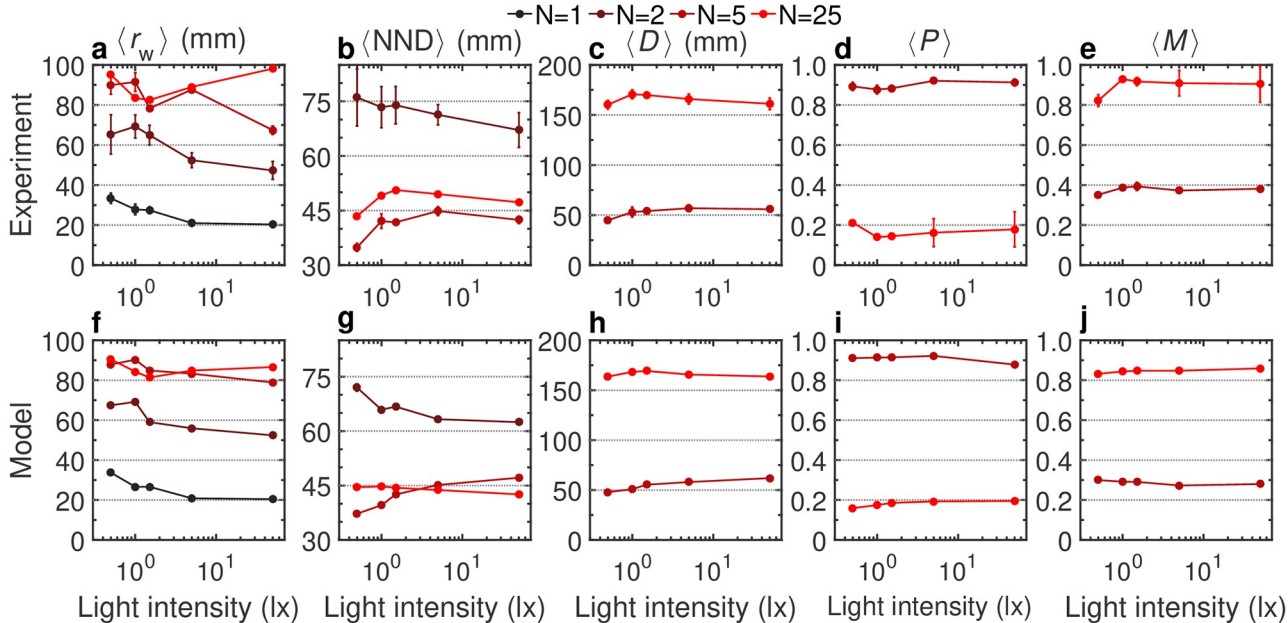

**Fig 10. Effect of light intensity on collective behavior for $N$ = 1, 2, 5, and 25 fish. a,f** Average distance of a fish to the wall $\langle r_w \rangle$. **b,g** Average distance of a fish to its nearest neighbor $\langle$NDD$\rangle$. **c,h** Average group dispersion $\langle D \rangle$. **d,i** Average group polarization $\langle P \rangle$. **e,j** Average group milling $\langle M \rangle$. **a-e** Experimental data, and **f-j** numerical simulations of the model, for different values of the light intensity (0.5, 1, 1.5, 5, and 50 lx) and different group sizes $N$ = 1, 2, 5, and 25 (from dark to light red). Gray dotted lines are a guide to the eye.

$\langle P \rangle \approx 0.9$, which is maximal at the highest illumination. The localization of the group and its strong polarization automatically results in a weak milling order parameter. Conversely, and as noted above, the groups of 25 fish are spread over the entire tank, while still concentrating near the wall. The spatial distribution of the group hence becomes rotationally invariant, leading to a negligible polarization. Yet, the group still presents a strong orientational order, with a large majority of fish turning around the tank in the same rotational direction, hence leading to a strong milling, with $\langle M \rangle \approx 0.9$.

Overall, Fig 10 shows that varying the light intensity has a greater impact when fish swims alone or in pairs than when they swim in larger groups.

The numerical simulations of the model including the interactions of the fish with their two most influential neighbors [60] and the modulation of social interactions with light intensity is in qualitative (S7 and S8 Videos) and fair quantitative agreement with the experimental results in both group sizes and different lighting conditions (Figs 10F–10J and S12 and S13). The parameter values used in the simulations of the model are listed in S7 (for $N$ = 5 fish) and S8 (for $N$ = 25 fish) Tables.

## Discussion

Understanding how environmental parameters impact both individual behavior and social interactions at the individual level, as well as the emergent properties observed at the collective level, is a fundamental question in studies of collective animal movements [10, 61, 62]. In fish, the level of illumination is an important ecological factor that strongly affects schooling behavior and regulates their behavioral activities [63, 64]. When illumination falls below a critical level, fish cannot perceive the visual information they need to coordinate their swimming [59]. Most studies on the effects of the level of illumination on schooling have only used descriptors, such as the swimming speed, the distribution of nearest neighbor distances or inter-individual

distances or the level of polarization and only a few of them have analyzed the behavioral responses at individual scale [43, 65].

In the present work, we measured and modeled the impact of the light intensity both at the individual scale (swimming behavior and social interactions between fish) and at the collective scale (collective motion patterns) in groups of *H. rhodostomus*. We used the procedure introduced by Calovi *et al.* [39] to extract the interactions between a fish and the wall, as well as with another fish, and to quantify the effects of light intensity on these interactions. Our results show that light intensity significantly alters the social interactions between individuals as well as the way fish swim and react to the obstacles such as the tank wall detected in their environment.

We find that, as light intensity decreases, the interactions between fish and between fish and the tank wall are weakened. Fish change their swimming direction more frequently and use smaller kick lengths, presumably to avoid collisions. Moreover, the intensity and the range of interactions with obstacles and neighboring fish increase with light intensity, the consequences being that fish move closer to each other and also closer to the wall when they swim in pairs. However, in larger groups of fish, as individuals only interact with their two most influential neighbors, only small groups of 5 fish remain cohesive and in these groups, resulting in a high polarization. In larger groups of 25 fish, the swimming patterns are no longer polarized, and the fish rotate around the center of the arena whatever the level of illumination with a high orientational milling order. This is mainly a consequence of the limited size of the tank and one can expect that without confinement, large groups would be much more polarized.

The measurements of the effects of light intensity on burst-and-coast swimming and individual interactions were then used to build and calibrate a model that quantitatively reproduces the dynamics of 1, 2, 5, and 25 fish, and the consequences of individual-level interactions on the spatial and angular distributions of the fish within the tank.

As observed in the experiments, the model shows that the coupling between wall avoidance behavior and the burst-and-coast motion results in the concentration of fish trajectories close to the wall that increases with light intensity. Remarkably, even if the repulsive interaction with the wall increases with light intensity, since the distance travelled by a fish between two kicks also increases, it moves more often close to the wall due to its burst-and-coast swimming mode [39]. Indeed, the longer the distance travelled between two kicks, the more difficult it is for a fish to escape the concave wall so that it spends more time close to the wall. Moreover, as a fish spends more time close to the wall, the intensity of the repulsive interaction must be higher. The model also correctly recovers the transition between a highly polarized group with a low milling rotational order for $N = 5$ fish, to a weak polarized group for $N = 25$ fish presenting a strong milling rotational order. The model also reproduces a similar average distance of fish to its nearest neighbor in both group sizes. As a consequence, the total space occupied by the group increases with group size, leading to a higher dispersion value. Finally, the model reproduces the fact that the influence of the light intensity on the different measured observables is more systematic and important for $N = 1$ and $N = 2$ fish than for larger groups of $N = 5$ and $N = 25$ fish. Our results suggest that there exist robust effective interactions between fish, since only the strength and range of these interactions but not their functional forms change with light intensity. This provides a general explanation for the way fish adapt their behavior and the way they interact with each other to environmental changes.

Overall, our approach, which combines experiments with data-driven computational modeling, has allowed us to decipher how the level of illumination affects the behavior and interactions among fish, and how the modulation of these interactions at the individual level leads to changes in collective movements observed at the group level. Our approach that leads to an explicit and predictive model can also be extended to understand and explain how the

modulation of social interactions and behavior by environmental parameters (e.g., light, temperature, flow speed, etc.) or physiological parameters (e.g., stress, hunger, etc.) affects collective behaviors in animal groups.

## Materials and methods

### Experimental procedures and data collection

**Ethics statement.** Experiments were approved by the Animal Experimentation Ethics Committee C2EA-01 of the Toulouse Biology Research Federation and were performed in an approved fish facility (A3155501) under permit APAFIS#27303-2020090219529069.v8 in agreement with the French legislation.

**Study species.** *Hemigrammus rhodostomus* (rummy-nose tetras) were purchased from Amazonie Labège in Toulouse, France. Fish were kept in 16 L aquariums on a 12:12 hour, dark:light photoperiod, at 24.9˚C (±0.8˚C) and were fed *ad libitum* with fish flakes. The average body length of the fish used in these experiments is 31 mm.

**Experimental setup.** We used a rectangular glass tank 120 × 120 cm supported by a 20 cm high metal beam structure. A circular arena of radius $R$ = 250 mm was set inside the experimental tank filled with 7 cm of water of controlled quality (50% of water purified by reverse osmosis and 50% of water treated by activated carbon) heated at 27.1˚C (±0.5˚C). The tank was surrounded by white opaque curtains and the experimental room was illuminated by four LED light panels proving homogeneous lighting. Light intensity was controlled through the control light panel of the room, allowing 5 illumination levels (0.5, 1, 1.5, 5, and 50 lx).

At the start of each trial, 1, 2, 5, or 25 fish were randomly removed from the breeding aquariums and placed in the circular arena. Fish were introduced in and acclimatized to the experimental tank and lighting conditions during a period of 10 minutes before the trial started. During each trial of one hour, individuals were swimming freely without external perturbation. Fish trajectories were recorded by a Sony HandyCam HD camera from above the setup at 25 Hz (25 frames per second) in HDTV resolution (1920 × 1080 p). Trials were carried out with 5 different light intensities (see S1–S4 Tables).

**Data extraction and pre-processing.** Positions of fish in each frame have been tracked with the tracking software FastTrack [66]. The FastTrack output format gives the position (in pixels) of each fish in each frame, with a time step of Δ = 0.04 s. Although the accuracy of the tracking was satisfactory, there were some identification errors, especially in large groups of fish. We corrected the wrong tracks by reassigning the identification numbers to the right fish. We used a sorting algorithm where the identities of the fish are sequentially reassigned so that the coordinates of each fish at the next time step are the closest to the coordinates they had at the previous frame. That is, the fish $i$ at time $t$ is assigned to the coordinates of fish $j$ at time $t + \Delta t$ such that the distance covered by the fish group is minimized. In the video, the diameter of the tank is about 1045 pixels, and the actual diameter is 500 mm. The conversion factor from pixels to meters is 0.478 mm/pix. In order to correct possible changes in the position of the experimental tank between two trials, the coordinate of the central pixel $(x_0, y_0)$ of the aquarium in the video must be modified on each of the videos.

The original pixel coordinates $(\hat{x}, \hat{y})$ are converted into metric coordinates so that the origin of the system of reference is located in the center of the tank:
$(x, y) = 0.478(\hat{x} - \hat{x}_0, \hat{y} - \hat{y}_0)$, where we have used the same conversion factor in both directions and $(\hat{x}_0, \hat{y}_0)$ are the pixel coordinates of the center of the tank.

During a test, it could happen that fish did not move. Therefore, we selected the phases during which a sustained swimming activity was observed. We considered that if the fastest fish in a group was swimming at a speed of less than 40 mm/s for more than 2 seconds, the fish were

stopping. The program extracted the sequences where the fish were active, removing stopping behavior. The data were then divided into several sequences of different duration for analysis. The proportion of time when individuals were detected active over the whole trial is listed in see S1–S4 Tables.

Fish trajectories were segmented according to the typical burst-and-coast swimming of this species. There is a succession of short acceleration phases called "kicks", during which a fish may change its direction of motion, followed by a phase of quasi-passive deceleration, during which the fish glides in an almost straight line until the next kick. The points of acceleration and gliding can be identified by the minimum and maximum values of the velocity time series. We use a Savitsky-Golay filter of degree three over a 0.36 s time window to smooth the raw time series, and divided the time series into accelerating and decelerating states. To limit noise, we merge any acceleration or deceleration periods smaller than 0.08 s. We assume that the times of the kicks coincide with the starting of the acceleration periods.

In [39], an almost perfect left/right symmetry was observed for single fish and pairs of fish. This means that a trajectory observed from the top of the tank has the same probability of occurring as the very same trajectory but as seen from under the tank (i.e., a "mirror trajectory"). Similarly, we consider that left/right symmetry exists in large groups ($N = 5, 25$). This reasonable assumption not only effectively doubles the data set, but also reduces the statistical uncertainty of the measured quantities and gives rise to more accurate interaction functions.

## Computational model

The duration of the bursting phase of *H. rhodostomus* (typically less than 0.1 s) is much smaller than that of the gliding phase (typically 0.5 s), and can thus be neglected. We also assume that fish choose their direction of motion at the kicking instant, maintaining their heading while decelerating in the gliding phase. The computational model thus consists of three equations per fish that determine the instant at which a kick takes place, and how the heading and position of the fish are updated at these kicking instants.

The $n$-th kick of fish $i$ starts at time $t_i^n$ and is characterized by a length $l_i^n$, a duration $\tau_i^n$, and the initial position and heading of the fish at the kicking instant, $\vec{u}_i^{\,n} = (x_i^n, y_i^n)$ and $\phi_i^n$ respectively. At the instant $t_i^n$, the fish chooses its new heading $\phi_i^{n+1}$ and moves along a straight segment of length $l_i^n$ during a time $\tau_i^n$, at the end of which it arrives at its new position $\vec{u}_i^{\,n+1}$, according to the following equations:

$$t_i^{n+1} = t_i^n + \tau_i^n, \tag{1}$$

$$\phi_i^{n+1} = \phi_i^n + \delta\phi_i^n, \tag{2}$$

$$\vec{u}_i^{\,n+1} = \vec{u}_i^{\,n} + l_i^n\, \vec{e}(\phi_i^{n+1}), \tag{3}$$

$\vec{e}(\phi_i^{n+1})$ is the unit vector along the angular direction $\phi_i^{n+1}$ and $\delta\phi_i^n$ is the change of heading from kick $n - 1$ to kick $n$. The length and the duration of a kick performed by a fish are independent of those from previous kicks, and also of the kicks of other fish. Thus, when swimming in groups, the kicks of different fish are asynchronous and not necessarily of the same length.

Individual fish make decisions at discrete times, at which the relative state of the $N - 1$ other fish must be known to evaluate the social interaction exerted on the focal fish. This information must be collected at a time that does not necessarily coincide with the kicking time of the other fish. The instantaneous speed of the fish decreases quasi-exponentially during the

kick with a decay time $\tau_0$ [39], as observed experimentally in the data under each light condition. The peak speed $v_n$, the kick duration $\tau_n$ and the kick length $l_n$ are linked by the following relation: $l_n = v_n \tau_0 [1 - \exp(-\tau_n/\tau_0)]$. Therefore, the instantaneous position of a fish at a time $\Delta t$ after its $n$th kick and before its next $(n + 1)$th kick, i.e., during the gliding phase, is given by

$$\vec{u}(t_i^n + \Delta t) = \vec{u}_i^n + l_i^n \frac{1 - \exp(-\Delta t/\tau_0)}{1 - \exp(-\tau_n/\tau_0)} \vec{e}(\phi_i^{n+1}). \tag{4}$$

We consider that the heading variation of individual fish $\delta\phi_i$ results from the additive combination of fish spontaneous behavior, physical constraints of the environment (obstacles), and social interactions with other fish:

$$\delta\phi_i = \delta\phi_i^R + \delta\phi_i^W + \delta\phi_i^S, \tag{5}$$

where indices R, W, and S stand for Random, Wall, and Social, respectively. Experiments performed with only one fish in the tank are devoted to identify the shape and intensity of the random spontaneous behavior of the fish $\delta\phi_i^R$ and of the interaction with the tank wall $\delta\phi_i^W$. Experiments with two fish allow to determine the pairwise interaction functions of attraction and alignment, and experiments in larger groups (here 5 and 25 individuals) serve to identify the neighbors to which an individual fish pays attention to update its heading. The experiments carried out in this work show that the same model structure, and especially, the same form of interaction functions, can be used for all light conditions.

When performing a kick, it may happen that the position of the fish calculated at the end of the kick is out of the tank. In that case, the kick is rejected and a new angle of spontaneous variation $\delta\phi_R$, a new kick length $l_i^n$, and a new kick duration $\tau_i^n$ are sampled from their distributions, until the final position at the end of the kick is inside the tank. Moreover, we consider that the fish keeps a distance of comfort $l_c$ to the wall, so that the kick is only accepted if

$$|\vec{u}(t_i^n) + (l_i^n + l_c)\vec{e}_i(\phi_i^n + \delta\phi_i^n)| < R, \tag{6}$$

where $R$ is the radius of the tank. If, after a large number of tries (up to 1000), the fish is still outside the tank, then a random number is sampled from the uniform distribution in $(-\pi, \pi)$ and assigned to $\delta\phi_i^R$ until Eq (6) is verified.

**Spontaneous heading change and interaction of fish with the wall.** Spontaneous heading change can be described by a Gaussian noise $\delta\phi^R = \gamma_R g$, where $\gamma_R$ is the intensity of these variations and $g$ is a Gaussian random variable of zero mean and unit variance. However, experiments show that, when fish swim close to the wall, the amplitude of their heading fluctuations is substantially reduced. We thus introduce a reduction factor that depends on the distance of the fish to the wall $r_w$, a characteristic distance to the wall $l_w$, and a modulation of this effect $\alpha \in (0, 1)$, so that

$$\delta\phi^R = \gamma_R \left(1 - \alpha \exp\left[-\left(\frac{r_w}{l_w}\right)^2\right]\right) g, \tag{7}$$

where all three parameters $\gamma_R$, $l_w$, and $\alpha$ are measured experimentally for each light condition.

The effect of the wall on heading variation is described by a function that depends only on the relative state of the fish with respect to the wall, $(r_w, \theta_w)$. We assume decoupled contributions of each variable,

$$\delta\phi^w(r_w, \theta_w) = f_w(r_w)O_w(\theta_w), \tag{8}$$

where $O_w$ is an odd function accounting for the fact that the fish turns with the same intensity

but in the opposite direction when the wall is on the right side of the fish ($\theta_w > 0$) or on the left side ($\theta_w < 0$), so that $O_w(-\theta_w) = -O_w(\theta_w)$.

By means of a reconstruction procedure introduced in [39, 42], we obtain analytical expressions of these functions for each light condition,

$$f_w(r_w) = \gamma_w \exp[-(r_w/l_w)^2], \tag{9}$$

$$O_w(\theta_w) \propto \sin(\theta_w)[1 + 0.8\cos(2\theta_w)], \tag{10}$$

where $\gamma_w$ is the intensity and $l_w$ is the range of the wall repulsion, and the angular function is normalized so that the mean of the squared function in $[-\pi, \pi]$ is equal to 1: $(1/2\pi)\int_{-\pi}^{\pi} O_w(\theta)^2 d\theta = 1$. The same normalization is applied for all angular functions in the model.

**Social interactions between fish.** Social interactions between two fish are described by pairwise functions of the relative state of the fish $(d, \psi, \Delta\phi)$, where $d$ is the distance between them, $\psi$ is the viewing angle with which the focal fish perceives the other fish, and $\Delta\phi = \phi_j - \phi_i$ is their relative alignment. The analysis of the experimental data shows that two distinct interactions are at play, attraction and alignment, so the social term is simply split into two additive parts:

$$\delta\phi^S(d, \psi, \Delta\phi) = \delta\phi_{Att}(d, \psi, \Delta\phi) + \delta\phi_{Ali}(d, \psi, \Delta\phi). \tag{11}$$

As for the function describing the effect of the wall, we assume that each state variable contributes to the heading change in a decoupled multiplicative form,

$$\delta\phi_{Att}(d, \psi, \Delta\phi) = f_{Att}(d)\, O_{Att}(\psi)\, E_{Att}(\Delta\phi), \tag{12}$$

$$\delta\phi_{Ali}(d, \psi, \Delta\phi) = f_{Ali}(d)\, O_{Ali}(\Delta\phi)\, E_{Ali}(\psi), \tag{13}$$

where $f$ denotes the strength of the interaction and $O$ and $E$ are respectively odd and even functions. The parity of the angular functions accounts for the intrinsic symmetry of each kind of interaction. For example, a fish is attracted by another fish with a force that has the same intensity but opposite sign if the other fish is at its right or left side, i.e., if $\psi < 0$ or $\psi > 0$ respectively. In turn, the force with which a fish attracts another has an intensity of the same sign, whatever the sign of the relative alignment $\Delta\phi$. Similarly, a fish tries to align with another fish with the same intensity independently of the side occupied by the other fish, but a fish turns right or left according to the sign of their relative alignment, $\Delta\phi > 0$ or $\Delta\phi < 0$, respectively.

By means of the already mentioned reconstruction procedure, we found the following analytical expressions of the pairwise social interaction functions:

$$f_{Att}(d) = \gamma_{Att}\frac{d/d_{Att} - 1}{1 + (d/l_{Att})^2}, \tag{14}$$

$$O_{Att}(\psi) \propto \sin(\psi)[1 - 0.33\cos(\psi)], \tag{15}$$

$$E_{Att}(\Delta\phi) \propto 1 - 0.6\cos(\Delta\phi) - 0.4\cos(2\Delta\phi), \tag{16}$$

$$f_{Ali}(d) = \gamma_{Ali}\frac{d}{d_{Ali}}\exp\left[-\left(\frac{d}{l_{Ali}}\right)^2\right], \tag{17}$$

$$O_{\text{Ali}}(\Delta\phi) \propto \sin(\Delta\phi)[1 + 0.33 \cos(\Delta\phi)], \tag{18}$$

$$E_{\text{Ali}}(\psi) \propto 1 + 1.18 \cos(\psi) - 0.49 \cos(2\psi). \tag{19}$$

The parameter $\gamma_{\text{Att}}$ (resp. $\gamma_{\text{Ali}}$) is the dimensionless intensity of the attraction (resp. alignment) interaction, $d_{\text{Att}}$ is the distance below which the attraction interaction changes sign (becoming repulsive), and $l_{\text{Att}}$ (resp. $\gamma_{\text{Ali}}$) is the range of interaction. $d_{\text{Ali}}$ is a fixed arbitrary scale set to ensure that $\gamma_{\text{Ali}}$ is dimensionless and to fix its typical magnitude. The values of these 6 parameters are extracted from the experiments by means of the reconstruction procedure for each light condition.

**Most influential neighbors.** In larger groups ($N > 2$), previous works have shown that *H. rhodostomus* can display the typical collective behavior of schooling and milling even when individual fish uses only the information about two specific neighbors [60, 67], selected according to the instantaneous *influence* they have on the heading variation of the focal fish. We define the influence $\mathcal{I}(d, \psi, \Delta\phi)$ that fish $j$ exerts on fish $i$ at time $t$ as the absolute value of the contribution of $j$ to the instantaneous heading change of $i$:

$$\mathcal{I}(d, \psi, \Delta\phi) = |\delta\phi^{\text{S}}(d, \psi, \Delta\phi)|. \tag{20}$$

We assume that fish combine the information about their neighbors in an additive form,

$$\delta\phi_i^{\text{S}} = \sum_{j=1}^{k} \delta\phi^{\text{S}}(d_{ij}, \psi_{ij}, \Delta\phi_{ij}), \tag{21}$$

where $k$ is the number of most influential neighbors taken into account. We take $k = 2$ for all light conditions, a value which was shown to lead to the best agreement with experiments in [60].

**Modulation of kick length and duration with the distance between fish.** Experimental data in all light conditions show that, when swimming in pairs, the kick length $l_i^n$ depends on the distance between fish $d$: the closer the fish, the shorter the kick (S9 Fig). We thus define a decoupled modulation function $F_{\text{m}}(d, \psi, \Delta\phi) = f_{\text{m}}(d)g_{\text{m}}(\psi)h_{\text{m}}(\Delta\phi)$ that depends on the relative state variables of pairs of fish and apply the same procedure of extraction used to build the social interaction functions, finding that the main contribution to kick length variation is due to the distance between fish, so that $g_{\text{m}}(\psi) \approx h_{\text{m}}(\Delta\phi) \approx 1$. The modulation function can then be written as

$$F_{\text{m}}(d) = l_{\text{m}} - \gamma_{\text{m}}(d + d_{\text{m}})\, e^{d/l_{\text{m}}}, \tag{22}$$

where $l_{\text{m}}$ is a saturation value at long distance, $\gamma_{\text{m}}$ is the intensity of the modulation, and $l_{\text{m}} - d_{\text{m}}$ is the distance for which the modulation is maximal.

## Quantification of collective behavior

The instantaneous state of the fish group can be characterized by means of three observables: dispersion, polarization, and milling.

The dispersion or radius of gyration of the group, $D(t) \in [0, R]$, is a measure of the total space occupied by the group, defined as

$$D(t) = \sqrt{\frac{1}{N}\sum_{i=1}^{N}\|\vec{u}_i - \vec{u}_B\|^2}, \tag{23}$$

where $\vec{u}_B = (x_B, y_B)$ is the position of the barycenter (center of mass) of the group, whose

velocity is given by $\vec{v}_B = (v_x^B, v_y^B)$, with

$$x_B = \frac{1}{N}\sum_{i=1}^{N} x_i(t), \quad v_x^B = \frac{1}{N}\sum_{i=1}^{N} v_x^i(t), \tag{24}$$

and similar expressions for $y_B$ and $v_y^B(t)$. The heading angle of the barycenter is given by its velocity vector, $\phi_B = \mathrm{ATAN2}(v_y^B(t), v_x^B(t))$. Low values of $D(t)$ correspond to highly cohesive groups, while high values of $D(t)$ imply that individuals are spatially dispersed.

The polarization $P(t) \in [0, 1]$ is a measure of the degree of alignment of the fish:

$$P(t) = \frac{1}{N}\left\|\sum_{i=1}^{N}\vec{e}_i(t)\right\|, \tag{25}$$

where $\vec{e}_i$ is the unit vector pointing in the fish heading direction. Polarization is high when $P$ is close to 1, meaning that the $N$ fish are aligned and point in the same direction. Polarization is low when the $N$ headings are weakly correlated, or even totally uncorrelated, then leading to the estimate $P \approx 1/\sqrt{N}$ resulting from the law of large numbers. Smaller values of $P$ require that the $N$ headings cancel each other, e.g., when directions are collinear and opposite.

The milling $M(t) \in [0, 1]$ measures how much the fish turn in the same direction around the center of the tank, independently of the direction of rotation. It is defined as

$$M(t) = \left|\frac{1}{N}\sum_{i=1}^{N}\sin\left(\bar{\theta}_w^i(t)\right)\right|, \tag{26}$$

where $\bar{\theta}_w^i(t) = \bar{\phi}_i - \bar{\theta}_i$. Variables with a bar are defined in the barycenter system of reference: $\bar{x}_i = x_i - x_B$, $\bar{v}_x^i = v_x^i - v_x^B$ (similar expressions for the $y$-components). Then, the angles of relative position and heading of fish $i$ with respect to $B$ are $\bar{\theta}_i = \mathrm{ATAN2}(\bar{y}_i, \bar{x}_i)$ and $\bar{\phi}_i = \mathrm{ATAN2}(\bar{v}_y^i, \bar{v}_x^i)$ respectively.

Other observables based on the barycenter, such as its distance to the wall $r_w^B = R - \sqrt{x_B^2 + y_B^2}$ and its angle of incidence $\theta_w^B = \mathrm{ATAN2}(v_y^B, v_x^B)$, can be used for small groups ($N = 5$). However, when $N = 25$ the fish occupy the whole tank almost uniformly and the barycenter's state is not informative.

## Parameter estimation and simulations

For each light condition, we performed 20 simulation runs with different initial conditions and with a duration of 1000 s.

When swimming alone, the kick length of the $n$-th kick of a fish is calculated from the peak speed $v_n$ and the kick duration $\tau_n$, sampled from bell-shaped distributions obtained in the experiment of each light condition. For example, in the 50 lx condition, we use $\tau_n = -0.5\,\bar{\tau}\ln(r_1 r_2)$, where $\bar{\tau} = 0.45$ s is the mean kick duration observed at 50 lx, and $r_1$ and $r_2$ are two uniform random numbers sampled in (0, 1). To perfectly fit the experimental curve, a new value is sampled each time that $\tau_n < 0.22$ s.

When swimming in groups, the kick length $l_n$ can depend on the distance $d$ between the focal fish and its most influential neighbor. Moreover, the modulation of kick length and duration with the distance between also depends on light intensity (S9 Fig). In that case, $l_n$ is sampled directly from the distribution whose mean is given by the modulation function: $l_n = -0.5\,f_m(d)\ln(r_1 r_2)$. We use this modulation when $N = 2$, but neglect it when $N \geq 5$, where the kick

length is again sampled from a random distribution with $f_m(d)$ replaced by the mean kick length.

All parameter values used in the simulations for all groups sizes and all light conditions are reported in see S5–S8 Tables.

## Supporting information

**S1 Table. List of experiments with one fish.**
(XLSX)

**S2 Table. List of experiments with two fish.**
(XLSX)

**S3 Table. List of experiments with 5 fish.**
(XLSX)

**S4 Table. List of experiments with 25 fish.**
(XLSX)

**S5 Table. Parameters used in the simulations for $N = 1$.**
(XLSX)

**S6 Table. Parameters used in the simulations for $N = 2$.**
(XLSX)

**S7 Table. Parameters used in the simulations for $N = 5$.**
(XLSX)

**S8 Table. Parameters used in the simulations for $N = 25$.**
(XLSX)

**S9 Table. Wilcoxon test for $N = 1$.** This test evaluates the p-value associated with the hypothesis that the location of data ($\tau$, $l$ or $v_0$) for 2 different lights are significantly different. We provide 2 significant digits after rounding, and an entry "0.00" indicates that the p-value is less than 0.01.
(XLSX)

**S10 Table. Wilcoxon test for $N = 2$.**
(XLSX)

**S11 Table. Wilcoxon test for $N = 5$.**
(XLSX)

**S12 Table. Wilcoxon test for $N = 25$.**
(XLSX)

**S1 Video. Effect of light intensity on individual swimming behavior in rummy-nose tetra (*Hemigrammus rhodostomus*).** Video excerpts of experiments with a single fish swimming alone in a circular tank of radius 250 mm under five different light intensities (0.5, 1, 1.5, 5, and 50 lx).
(MP4)

**S2 Video. Effect of light intensity on social interactions between two fish in rummy-nose tetra (*Hemigrammus rhodostomus*).** Video excerpts of experiments with 2 fish swimming in a circular tank of radius 250 mm under five different light intensities (0.5, 1, 1.5, 5, and 50 lx).
(MP4)

**S3 Video. Effect of light intensity on collective behavior in groups of 5 fish in rummy-nose tetra (*Hemigrammus rhodostomus*).** Video excerpts of experiments with a group of 5 fish swimming in a circular tank of radius 250 mm under five different light intensities (0.5, 1, 1.5, 5, and 50 lx).
(MP4)

**S4 Video. Effect of light intensity on collective behavior in groups of 25 fish in rummy-nose tetra (*Hemigrammus rhodostomus*).** Video excerpts of experiments with a group of 25 fish swimming in a circular tank of radius 250 mm under five different light intensities (0.5, 1, 1.5, 5, and 50 lx).
(MP4)

**S5 Video. Numerical simulations of the model with a single fish under different light intensities.** Representative example of a simulation of a single fish swimming in a circular tank of radius 250 mm under five different light intensities (0.5, 1, 1.5, 5, and 50 lx). The size of the simulated fish does not correspond to the actual dimensions of the real fish and is used for ease of visualization.
(MP4)

**S6 Video. Numerical simulations of the model with 2 fish under different light intensities.** Representative example of a simulation of 2 fish swimming in a circular tank of radius 250 mm under five different light intensities (0.5, 1, 1.5, 5, and 50 lx). The size of the simulated fish does not correspond to the actual dimensions of the real fish and is used for ease of visualization.
(MP4)

**S7 Video. Numerical simulations of the model with a group of 5 fish under different light intensities.** Representative example of a simulation of a group of 5 fish swimming in a circular tank of radius 250 mm under five different light intensities (0.5, 1, 1.5, 5, and 50 lx). Each fish interacts with its two most influential neighbors. The size of the simulated fish does not correspond to the actual dimensions of the real fish and is used for ease of visualization.
(MP4)

**S8 Video. Numerical simulations of the model with a group of 25 fish under different light intensities.** Representative example of a simulation of a group of 25 fish swimming in a circular tank of radius 250 mm under five different light intensities (0.5, 1, 1.5, 5, and 50 lx). Each fish interacts with its two most influential neighbors. The size of the simulated fish does not correspond to the actual dimensions of the real fish and is used for ease of visualization.
(MP4)

**S1 Fig. Burst-and-coast motion of fish swimming alone in the tank.** Time series of the instantaneous speed of one fish under different light intensities: **a** 0.5, **b** 1, **c** 1.5, **d** 5 and **e** 50 lx. Colored vertical lines represent local minima (red) and maxima (blue) of the speed. Time intervals going from a red line to the next blue line correspond to the bursting acceleration phase, intervals going from a blue line to the next red line correspond to the decelerating gliding phase.
(PDF)

**S2 Fig. Normalized average decay of fish speed right after a kick when the fish swims alone ($N = 1$). a** Exponential deceleration during the gliding phase averaged along all kicks and normalized with the value of the speed at the kicking instant, for different light intensities 0, 0.5, 1, 5, and 50 lx (from dark to light blue). Wide solid lines are experimental measures, dashed lines

are exponential approximations of the form $\exp(-t/\tau_0)$, where $\tau_0$ is the relaxation time: $\tau_0 \approx$ 0.34 (0.5 lx), 0.66 (1 lx), 0.76 (1.5 lx), 0.83 (5 lx), 0.87 (50 lx). **b** Mean relaxation time $\tau_0$ as a function of the light intensity (black circles). The red dashed line shows the trend of the average value with the light intensity.
(PDF)

**S3 Fig. Effects of light intensity on the spontaneous heading change of a fish swimming alone ($N$ = 1).** Probability density function (PDF) of the angle variation $\delta\phi$ when the fish is far from the wall ($r_w > 60$ mm) in five different light intensities: 0.5, 1, 1.5, 5 and 50 lx (from dark to light blue). Colored dots: measures from the experiments. Dashed lines: approximation with Gaussian distributions, with $\gamma_R$=0.26, 0.34, 0.40, 0.42, and 0.43 respectively.
(PDF)

**S4 Fig. Effect of light intensity on the fish interaction with the tank wall when the fish swims alone ($N$ = 1).** Function of repulsion $f_w(r_w)O_w(\theta_w)$ as extracted from the experiments by means of the reconstruction procedure (dots), and analytical approximations used in the numerical simulations (solid lines), for different light intensities: 0.5, 1, 1.5, 5, and 50 lx (from dark to light blue). **a-e** Intensity of the interaction $f_w(r_w)$ as a function of the fish distance to the wall $r_w$. **f-j** Intensity of the interaction $O_w(\theta_w)$ as a function of the relative orientation of the fish to the wall $\theta_w$. Orange lines correspond to the analytical approximation of a single discrete function combining all light conditions: $O_w(\theta_w) = 1.9612 \sin(\theta_w)[1 + 0.8 \cos(2\theta_w)]$.
(PDF)

**S5 Fig. Effect of the kick length $l$ on the spatial distribution of a fish swimming alone ($N$ = 1). a** Probability density function (PDF) of the distance of the fish to the wall $r_w$ as a function of light intensity when only the kick length $l$ is changed in the model. **b** Schematic diagram of the motion of a single fish between two kicks. The distance travelled by the fish between two kicks is greater at 50 lx than at 0.5 lx; as a consequence, the fish moves closer to the wall when the light intensity is higher.
(PDF)

**S6 Fig. Normalized average decay of fish speed right after a kick when fish swim in pairs ($N$ = 2). a** Exponential deceleration during the gliding phase averaged along all kicks and normalized with the value of the speed at the kicking instant, for different light intensities 0, 0.5, 1, 5, and 50 lx (from dark to light blue). Wide solid lines are experimental measures, dashed lines are exponential approximations of the form $\exp(-t/\tau_0)$, where $\tau_0$ is the relaxation time: $\tau_0 \approx$ 0.39 (0.5 lx), 0.63 (1 lx), 0.69 (1.5 lx), 0.71 (5 lx), 0.79 (50 lx). **b** Mean relaxation time $\tau_0$ as a function of the light intensity (black circles). The red dashed line shows the trend of the average value with the light intensity.
(PDF)

**S7 Fig. Effects of light intensity on the attraction interaction between two fish ($N$ = 2).** Components of the attraction interaction function $f_{Att}(d)$, $O_{Att}(\psi)$, and $E_{Att}(\Delta\phi)$ as functions of the distance between fish $d$, the viewing angle $\psi$, and the relative heading $\Delta\phi$, for different light intensities: **a-c** 0.5 lx, **d-f** 1 lx, **g-i** 1.5 lx, **j-l** 5 lx, and **m-o** 50 lx (from dark to light blue). Color dots correspond to the discrete values resulting from the reconstruction procedure, extracted from the experimental data of the corresponding intensity of light. Solid lines correspond to the analytical approximation of the discrete function. Orange lines correspond to the analytical approximation of a single discrete function combining all light conditions.
(PDF)

**S8 Fig. Effects of light intensity on the alignment interaction between two fish ($N = 2$).** Components of the attraction interaction function $f_{\mathrm{Ali}}(d)$, $E_{\mathrm{Ali}}(\psi)$, and $O_{\mathrm{Ali}}(\Delta\phi)$ as functions of the distance between fish $d$, the viewing angle $\psi$, and the relative heading $\Delta\phi$, for different light intensities: **a-c** 0.5 lx, **d-f** 1 lx, **g-i** 1.5 lx, **j-l** 5 lx, and **m-o** 50 lx (from dark to light blue). Color dots correspond to the discrete values resulting from the reconstruction procedure, extracted from the experimental data of the corresponding intensity of light. Solid lines correspond to the analytical approximation of the discrete function. Orange lines correspond to the analytical approximation of a single discrete function combining all light conditions.
(PDF)

**S9 Fig. Effect of light intensity on the modulation of the kick length with the distance between fish.** Modulation function ($F_{\mathrm{m}}(d)$) of the mean value used in the distribution from which kick lengths are sampled, as a function of the distance between fish $d$, and for different light intensities: 0.5, 1, 1.5, 5, and 50 lx (from dark to light blue). Dots correspond to the discrete functions resulting from the reconstruction procedure and extracted from the experimental data. Solid lines correspond to the smooth analytical approximations of these discrete functions.
(PDF)

**S10 Fig. Effects of light intensity on burst-and-coast swimming in groups of 5 fish. a-c** Probability density function (PDF) of kick duration $\tau$, kick length $l$, and peak speed $v_0$ respectively, at different light intensities: 0.5, 1, 1.5, 5 and 50 lx (from dark to light blue). **d-f** Average value of kick duration $\langle\tau\rangle$, kick length $\langle l\rangle$, and peak speed $\langle v_0\rangle$ respectively, at different light intensities. Solid circles are the average values on all experiments; error bars represent the standard error. Red dashed lines show the trend of the average value with the light intensity.
(PDF)

**S11 Fig. Effects of light intensity on burst-and-coast swimming in groups of 25 fish. a-c** Probability density function (PDF) of kick duration $\tau$, kick length $l$, and peak speed $v_0$ respectively, at different light intensities: 0.5, 1, 1.5, 5 and 50 lx (from dark to light blue). **d-f** Average value of kick duration $\langle\tau\rangle$, kick length $\langle l\rangle$, and peak speed $\langle v_0\rangle$ respectively, at different light intensities. Solid circles are the average values on all experiments; error bars represent the standard error. Red dashed lines show the trend of the average value with the light intensity.
(PDF)

**S12 Fig. Quantification of collective behavior in groups of 5 fish.** Probability density functions (PDF) of **a,d** the distance to the wall $r_{\mathrm{w}}$, **b,e** the relative angle to the wall $\theta_{\mathrm{w}}$, **c,f** the distance to the nearest neighbor NND, **g,j** dispersion $D$, **h,k** polarization $P$, and **i,l** milling $M$, for five different light intensities 0.5, 1, 1.5, 5, and 50 lx (from dark to light blue). Solid lines (**a-c, g-i**) correspond to experimental measures, dashed lines (**d-f, j-l**) to numerical simulations of the model.
(PDF)

**S13 Fig. Quantification of collective behavior in groups of 25 fish.** Probability density functions (PDF) of **a,d** the distance to the wall $r_{\mathrm{w}}$, **b,e** the relative angle to the wall $\theta_{\mathrm{w}}$, **c,f** the distance to the nearest neighbor NND, **g,j** dispersion $D$, **h,k** polarization $P$, and **i,l** milling $M$, for five different light intensities 0.5, 1, 1.5, 5, and 50 lx (from dark to light blue). Solid lines (**a-c, g-i**) correspond to experimental measures, dashed lines (**d-f, j-l**) to numerical simulations of the model.
(PDF)

## Author Contributions

**Conceptualization:** Guy Theraulaz.

**Formal analysis:** Tingting Xue, Xu Li, GuoZheng Lin, Ramón Escobedo, Clément Sire.

**Funding acquisition:** Zhangang Han, Xiaosong Chen, Guy Theraulaz.

**Investigation:** Tingting Xue, Xu Li, GuoZheng Lin.

**Methodology:** Ramón Escobedo, Guy Theraulaz.

**Project administration:** Guy Theraulaz.

**Software:** Tingting Xue, Xu Li.

**Supervision:** Zhangang Han, Guy Theraulaz.

**Validation:** Clément Sire, Guy Theraulaz.

**Visualization:** Xu Li.

**Writing – original draft:** Tingting Xue, Xu Li, Guy Theraulaz.

**Writing – review & editing:** Ramón Escobedo, Clément Sire.

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
