## [Decision Letter · Decision Letter 0]

13 Sep 2023

Dear Dr. Theraulaz,

Thank you very much for submitting your manuscript "Tuning social interactions’ strength drives collective response to light intensity in schooling fish" for consideration at PLOS Computational Biology. As with all papers reviewed by the journal, your manuscript was reviewed by members of the editorial board and by several independent reviewers. The reviewers appreciated the attention to an important topic. Based on the reviews, we are likely to accept this manuscript for publication, providing that you modify the manuscript according to the review recommendations.

Sincerely,

Feng Fu

Academic Editor

PLOS Computational Biology

James O'Dwyer

Section Editor

PLOS Computational Biology

Reviewer's Responses to Questions

**Comments to the Authors:**

Reviewer #1: Here the authors have performed a valuable study that helps reveal the influence of light illumination on schooling fish. By disabling the visual system in individual fish and as a group, the authors have found that light intensity impacts fish at both the individual scale and collective scale. Interestingly, fish exhibit changes in swimming modes with changes to light intensity, which was further modeled to show predictive patterns with group sizes. The authors show that light intensity plays a large role in collective swimming and that behaviors on the individual and group scale deteriorates as the visual information with their environment decreases. Overall, the manuscript is well-written, the figures are exceptional, and the modeling work in the study are novel and improve our understanding of was how fish school, which has been a tricky behavior to study. I only suggest that the authors better introduce the study and add additional discussion of a few results.

My specific minor concerns are itemized below. The greatest of these lies in the lack of a presented motivation for this work. In the abstract and introduction, the authors should articulate how the present work makes a novel contribution to this area of investigation.

Specific suggestions:

L13:L25 – Perhaps emphasize how the current focus differs from previous papers; therefore, establishing that computational modeling can be used to predict sensory-based collective behaviors. What makes this study slightly novel and exciting is the modeling of visual cues to predict collective motion. As such, I suggest adding a sentence referring to how this study uses modeling as a way to test the visual system of schooling fish, which makes it different from other fish schooling papers.

L26 – We hypothesize that the behavior might be multisensory, including the lateral line and vision. There has been minimal work in whether fish can use other sensory cues such as the vestibular system, separate from the lateral line, or olfaction, as seen in Pacific Salmon as a homing cue. Here, I would also cite: Mckee et al. (2020) and Mekdara et al. (2018) as they discuss that the behavior is most likely multisensory and can be species specific.

L61 – After the previous paragraph, it is not clear exactly how the present study builds upon previous work. Though it’s been clearly written in the discussion, I believe a few sentences justifying the importance of the present study, and explicitly stating how the study differs from others, would be useful here.

L70:L81 & Fig. 2, 6, etc. – Were basic statistical tests for averages not done for different light conditions? Although probability density function plots and fitted curves are sufficient to show changes in data trends, effect size or how large the changes compare between groups has some value for biological significance. This is a suggestion as it doesn’t change the scope/results of the paper.

L184:L187 – Do you mean that there is a tank size effect for large groups?

Fig. 10 – I think the model being closely aligned to the experimental data is quite interesting here, especially with the polarization and milling parameters. How does the results of the model reflect the natural species-specific schooling traits? For example, what are the natural group sizes of adult H. rhodostomus and does the model reflect that?

L248 – As mentioned in the above comment, is the hypothesis here that large groups no longer care about directional swimming, either with wall attraction/edge-fixation, because of the lack of motivation or is the dynamic different in larger groups? Or was it simply that the tank was too small, which reduces the polarization? Were there any flow cues added to the tank? The assumption here is that H. rhodostomus is a highly visual schooling-based species; and that once in larger groups, the animals no longer want to swim unidirectionally following an edge cue (the wall), especially if the visual information becomes noisier. How does the visual information change in larger groups or does it change at all since it seems that a focal fish only tracks one or two fish at a time?

L268 – There isn’t a large explanation as to why the group “dispersion” parameter could be so different between n = 5 fish versus n = 25 fish. Is n = 5 fish the optimal group size under experimental conditions? What would be the optimal group size as a function of polarization and/or milling before a dramatic decrease in these parameters?

L273:276 – The conclusion of the paper falls a little short as the authors make very little inferences to explain how the computation models can help predict and test the biological mechanism or the functional aspects of the system. The scope of the paper is mainly data-driven with small inferences to biology. I would like to encourage the authors to add a bit of biological context as to how the computational model can be used to examine other modes of collective behaviors seen in other species.

Reviewer #2: The authors present a systematic investigation of the impact of light intensity on the social interaction of fish by explicitly also taking into account the change in the swimming behavior of individual fish. Light intensity is the most obvious factor modulating visual perception in fish, thus it is natural to assume that it has an important impact on social interactions. Thus many studies investigated the role of light intensity, they typically were restricted to quantification of fish behavior and structure of the school without directly aiming at quantifying social interactions. In general, it is intuitive to assume that decreased light intensity, leads to weaker social interactions mediated by vision, and the previous observations seem to point into this direction. Here, the main result of the present paper, which explicitly confirms this is not really new. What is novel are the computational methodological aspects, of explicitly mapping out and fitting the interaction functions for the various contributions governing fish swimming behavior (interaction with the wall, and social interactions attraction+alignment) at different light intensities, using a general methodology established previously by the same group. What the results show that the functional shape of the interactions does not change with the light intensity, which in principle could also be the case. To my knowledge this is the most in-depth and systematic investigation of these aspects up to date, and thus of potential interest to a broader audience interested in collective behavior, fish behavior and visual ecology more generally. Importantly, the authors combine their experimental work with simulations of an individual-based model to test how far the observed changes in social interactions, can explain also the emergent patterns of collective swimming.

The paper is well written, the figures are of good quality thus, I think the work if potentially suitable for publication with PLoS Comp Biol. However, there is a number of aspects, I would expect the authors to consider / revise.

1) Individual swimming:

- The authors state that the spontaneous heading change away from walls is Gaussian distributed. While this seems to be (at least approx.) the case for the larger light intensities. The results in Supp. Fig 3, clearly show deviations from a Gaussian for smaller intensities (<=1.5lx). Here it seems to be more exponentially distributed. I expect this to be correctly reported and maybe also discussed in the context of other experimental findings.

The caption of Supp Fig 3 it states that solid lines are the approximation with a Gaussian distribution. However, the lines are dashed.

2) Interactions with the wall:

For the largest light intensities 5.0 and 50lx there appears to be clearly an attractive force to the wall above 70mm. Which the authors seem not to comment on / discuss.

Furthermore, what is interesting is that also the fit of the angular dependence seems to become worst at maximal light intensity of 50lx. I think this is at least worth to mention as well as well as potentially discuss.

3) Discussion general findings: I think the result that the functional form of the interactions does not change with the light intensity, should be highlighted more. As this is not trivial, as in principle the functional form could also change due to constraints in vision. The results suggest that there is somehow a robust effective interaction between the fish, where only the strength is modulated.

**Have the authors made all data and (if applicable) computational code underlying the findings in their manuscript fully available?**

Reviewer #1: Yes

Reviewer #2: Yes

PLOS authors have the option to publish the peer review history of their article (what does this mean?). If published, this will include your full peer review and any attached files.

Reviewer #1: No

Reviewer #2: No

Figure Files:

Data Requirements:

Reproducibility:

References:

---

## [Decision Letter · Decision Letter 1]

26 Oct 2023

Dear Dr. Theraulaz,

We are pleased to inform you that your manuscript 'Tuning social interactions’ strength drives collective response to light intensity in schooling fish' has been provisionally accepted for publication in PLOS Computational Biology.

Best regards,

Feng Fu

Academic Editor

PLOS Computational Biology

James O'Dwyer

Section Editor

PLOS Computational Biology

Reviewer's Responses to Questions

**Comments to the Authors:**

Reviewer #1: Well done. The paper is well written, the figures are of great quality, and I think the manuscript is suitable for publication with PLoS Comp Biol.

**Have the authors made all data and (if applicable) computational code underlying the findings in their manuscript fully available?**

Reviewer #1: Yes

PLOS authors have the option to publish the peer review history of their article (what does this mean?). If published, this will include your full peer review and any attached files.

Reviewer #1: No

---

## [Editor Report · Acceptance letter]

9 Nov 2023

PCOMPBIOL-D-23-00863R1 

Tuning social interactions’ strength drives collective response to light intensity in schooling fish

Dear Dr Theraulaz,

I am pleased to inform you that your manuscript has been formally accepted for publication in PLOS Computational Biology. Your manuscript is now with our production department and you will be notified of the publication date in due course.

With kind regards,

Zsofi Zombor
